# Toxins of Okadaic Acid-Group Increase Malignant Properties in Cells of Colon Cancer

**DOI:** 10.3390/toxins12030179

**Published:** 2020-03-13

**Authors:** Danae Jiménez-Cárcamo, Carlos García, Héctor R. Contreras

**Affiliations:** 1Department of Basic and Clinical Oncology, Faculty of Medicine, University of Chile, Santiago 8380453, Chile; danae.jimenez.carcamo@gmail.com; 2Physiology and Biophysics Program, ICBM, Faculty of Medicine, University of Chile, Santiago 8380453, Chile

**Keywords:** diarrhetic shellfish poisoning (DSP), okadaic acid (OA), dinophysistoxin-1 (DTX-1), dinophysistoxin-2 (DTX-2), colorectal cancer

## Abstract

Diarrhetic shellfish poisoning (DSP) is a syndrome caused by the intake of shellfish contaminated with a group of lipophilic and thermostable toxins, which consists of okadaic acid (OA), dinophysistoxin-1 (DTX-1) and dinophysistoxin-2 (DTX-2). These toxins are potent protein Ser/Thr phosphatase inhibitors, mainly type 1 protein phosphatase (PP1) and type 2A protein phosphatase (PP2A). Different effects have been reported at the cellular, molecular and genetic levels. In this study, changes in cell survival and cell mobility induced by OA, DTX-1 and DTX-2 were determined in epithelial cell lines of the colon and colon cancer. The cell viability results showed that tumoral cell lines were more resistant to toxins than the nontumoral cell line. The results of the functional assays for testing cell migration, evaluation of cell death and the expression of proteins associated with cell adhesion showed a dual effect of toxins since in the nontumoral cell line, a greater induction of cell death, presumably by anoikis, was detected. In the tumoral cell lines, there was an induction of a more aggressive phenotype characterized by increased resistance to toxins, increased migration and increased FAK activation. In tumoral cell lines of colon cancer, OA, DTX-1/DTX-2 induce a more aggressive phenotype.

## 1. Introduction

Harmful algal blooms (HABs) are produced by microalgae, which under certain natural environmental and anthropogenic conditions increase in density in the sea. HABs can affect marine fauna, mammals, people and aquaculture, causing damage at economic, ecological and sanitary levels [1]. Diarrhetic shellfish poisoning (DSP) is a syndrome caused by the ingestion of contaminated shellfish, which is developed between 30 min and 6 h after ingestion, producing symptoms such as diarrhea (60%), nausea (46%), vomiting (31%) and abdominal pain (77%) [2,3]; in addition, it causes loss of epithelial integrity, erosion and intestinal hypersecretion, and increases paracellular permeability [4]. 

Okadaic acid (OA) is the main analog of DSP toxins. Other toxins of this group are dinophysistoxin-1 (DTX-1), dinophysistoxin-2 (DTX-2) and acylated toxins in the C7 hydroxyl group, with a variable length chain of fatty acids which are called acyl derivatives [5]. These toxins are lipophilic in nature and stable to thermal variants [1]. Toxins of the OA-group are potent protein Ser/Thr phosphatase inhibitors, which produce hyperphosphorylation in cellular proteins and deregulation of different processes in cells. The main molecular targets of this group of toxins are type 1 protein phosphatase (PP1) and type 2A protein phosphatase (PP2A) [6]. The acute toxicity obtained *in vivo* by intraperitoneal injection in mice caused lethal dose 50 (LD_50_) values similar for DTX-1 and OA, while DTX-2 has a toxicity factor of 0.6 in relation to the other analogs (TEF, toxic equivalent factor) [7]. However, some studies argue that DTX-1 is more toxic than OA, considering its ability to destroy the integrity of epithelial cells and to produce apoptosis [8]. The differences in the toxicity of this group of toxins lie in the stereochemistry of the compounds since methylation in C31 and C35 modify its interaction with the binding sites of different phosphatases (PP1, PP2A and to a lesser extent PP5) [9]. Although most effects of this group of toxins are related to the inhibition of protein phosphatases, there are reports of other possible targets of action relevant to the toxicity of toxins of the OA-group [10]. In this sense, Louzao et al. [11] report that doses of 50–100 nM of OA in SH-SY5Y neuroblastoma cells cause decreased cellular expression of neuropeptide Y, which has a protective effect against diarrhea. Furthermore, Espiña et al. [12] reported that methyl okadaate, an artificial derivative of OA, is able to disaggregate the actin cytoskeleton and is able to decrease the metabolic expenditure in normal and immortalized rat hepatocytes by an independent route to the inhibition of PP1 and PP2A. The importance of studying the toxicity and mechanisms of action of OA and its analogs in intestinal epithelial cells is because the gastrointestinal tract is their main target of action, and the symptoms produced by DSP syndrome mainly affect the digestive system. However, it has been determined that after high concentrations of OA, corresponding to acute poisoning, toxins pass through the bloodstream to other tissues, producing ulcers, hyperplasia, necrosis and atrophy [4].

There are studies that statistically correlate shellfish consumption with the incidence of colorectal cancer in the Spanish population. A seven-fold increase in shellfish consumption was associated with a two-fold increased risk of developing colorectal cancer [13]. Another study suggests there is an association between the exposure to toxins of the OA-group and the risk of cancer in the gastrointestinal tract, colon and stomach, mainly in the population of the French coast [14]. To study the relationship between cancer and toxins of the OA-group, Valdiglesias et al. [15] studied the expression of 10 genes related to carcinogenic processes in SH-SY5Y neuronal cells exposed to OA, in which alterations in the pattern of expression were obtained, confirming a possible association between the consumption of contaminated shellfish and the incidence of cancer in humans. 

The identification of risk factors is important for the prevention and incidence of cancer and, although there are no studies on the actual risk of consumption of shellfish contaminated with DSP toxins to date, the findings indicate that toxins in the OA-group behave as tumorigenic agents. The objective of this work is to determine the changes induced by okadaic acid, DTX-1 and DTX-2 in cell survival and mobility, in the colon cancer cell lines and a nontumoral colon epithelial cell line.

## 2. Results

### 2.1. Cell Viability

CCD 841 CoN, Sw480 and Sw620 cells were exposed to OA, DTX-1 and DTX-2 toxins for 12 h. The three toxins show a dose-dependent effect on cell lines (Table 1). In CCD 841 CoN cells, the IC_50_ value for OA was 54.4 nM, the IC_50_ value for DTX-1 was 43.5 nM and the IC_50_ value for DTX-2 was 81.2 nM. In Sw480 cells, the IC_50_ value for OA was 89.1 nM, the IC_50_ value for DTX-1 was 113.8 nM and the IC_50_ value for DTX-2 was 187.2 nM. In Sw620 cells, the IC_50_ value for OA was 137.8 nM, the IC_50_ value for DTX-1 was 192.9 nM and the IC_50_ value for DTX-2 was 202.9 nM. 

### 2.2. PP2A Inhibition

In order to check the effect on PP2A, which is the main target of toxins belonging to the OA-group, the PP2A inhibition assay was performed. According to the results obtained, when cell lines were treated with OA, it was found that PP2A activity was significantly decreased only in tumoral cell lines (CCD 841 CoN 93.4%; Sw480, 36%; Sw620, 57.2%) (Figure 1). When cell lines were treated with DTX-1, a significant decrease in the PP2A activity was observed in the three cell lines (CCD 841 CoN, 24.6%; Sw480, 67.8%; Sw620, 50.7%), and when cell lines were treated with DTX-2, it was evidenced that the decreased PP2A activity was lower for tumoral cell lines (Sw480, 90% and Sw620, 66%), in contrast to the increased activity for the nontumoral cell line (CCD 841 CoN, 114.5%).

### 2.3. Cell Proliferation

As a complement to the results of MTT, the measurement of PCNA was performed by Western blotting, using the concentrations obtained by the MTT assay for 12h. The results of the PCNA protein expression show that no statistically significant variations were shown for the applied toxin doses and incubation time (Figure 2)

### 2.4. Evaluation of Cell Death

To evaluate the type of cell death, lactate dehydrogenase (LDH) assays were performed to assess the caspase-3/7 activity for apoptosis detection, and the annexin V/PI assay by flow cytometry to correlate both and apoptosis. The results showed decreased LDH release in all samples, both treated and nontreated samples (Figure 3). The values were close to 30% with respect to the maximum control activity of the kit (OA: CCD 841 CoN 30.4%, Sw480 31.5%, Sw620 26.5%; DTX-1: CCD 841 CoN 32.5%, Sw480 30%, Sw620 26.8%; DTX-2: CCD 841 CoN 30.3%, Sw480 29.9%, Sw620 28.1%). There was decreased LDH activity in tumoral cell lines after treatment with toxins, a trend that becomes statistically significant for the Sw620 cell lines treated with OA and DTX-1 and for the Sw480 cell lines treated with DTX-2. Regarding the activity of activated caspases-3/7, the results showed significant increases in tumoral cell lines (Sw480 and Sw620) after the treatments with OA and DTX-2 (Figure 4).

As observed in Figure 5, the V/PI assay using flow cytometry showed that in the controls with no toxin there is a very low percentage of cells in an active state of early process of apoptosis (annexin V (+) and PI (−)), late apoptosis (annexin V (+) and PI (+)) and necrosis (annexin V (−) and PI (+)). After treatments on all cell lines, no statistically significant changes in the percentage of apoptotic (late and early) cells were observed. After treatment with OA, only a significant increase in Sw620 cells was observed.

### 2.5. Migration Assay

To assess how toxins affect cell mobility, a chamber-based transwell migration assay was performed. According to the results, the migratory capacity increased significantly in tumoral cell lines after exposure to DTX-1. As shown in Figure 6, the Sw620 cell line increased the migration after treatment with the three toxins, which could indicate the induction of a more aggressive toxins’ phenotype.

### 2.6. FAK Expression

Given that FAK is an important kinase in cell mobility and morphology and its expression has been studied in other lines after exposure to OA, the FAK expression was assessed using Western blot and RT-pPCR. Regarding the protein expression of FAK and pFAK (Figure 7), a significant increase was observed in the Sw480 cell line after treatment with DTX-1, which was only statistically significant for pFAK397. In the treatment with DTX-2, a statistically significant increase was observed for pFAK397 and pFAK576/577. For Sw620 cell line, a significant increase was observed for pFAK397 after treatment with DTX-1. Regarding the mRNA expression of FAK in CCD 841 CoN cells, a statistically significant decrease was observed (>50%) in the cells treated if compared with the controls with no toxin (Figure 8). Moreover, a significant decrease was observed in Sw480 cells only after treatment with DTX-1. 

## 3. Discussion

OA, DTX-1 and DTX-2 are lipophilic toxins produced by dinoflagellates with different levels of toxicity; also, they are the main compounds associated with episodes of DSP [16]. The cell viability results obtained using the MTT assay showed that DTX-2 is less toxic than OA and DTX-1 when the values of IC_50_ were compared; this is in line with what has been reported previously, where OA and DTX-1 are considered to be equally toxic, while DTX-2 is almost 40% less toxic [17]. However, other studies have reported that DTX-1 is more toxic than OA and DTX-2 [9]. When comparing nontumoral cell lines with tumoral cell lines, the nontumoral cell line CCD 841 CoN proved to be much more sensitive to the effect of the three toxins, obtaining lower values of IC_50_ compared with Sw480 and Sw620 cell lines. Notwithstanding, it is known that the toxicity of this group of compounds varies between different cell lines [10], where the greater resistance of the tumoral cell lines suggests that cell death is not activated during exposure to toxins, and therefore, they are more exposed to long-term toxic effects, such as the induction of a more aggressive phenotype.

Regarding the morphological changes observed with the toxins after incubation, the effect is much more evident in the nontumoral cell lines; cells become spherical and tend to cluster, which suggests a loss of adhesion to the substrate (Appendix A). These observed changes may be due to the reorganization of the cytoskeleton, and they have been reported in different cell systems after treatment with OA. Some examples are those described in MCF-7 and Hela S3 cells treated with OA [11], as well as in SH-EP cells treated for 4 h with 250 nM of OA [18]. Valdiglesias et al. [13] reported morphological changes induced by OA in three different cell types: peripheral blood leukocytes, HepG2 hepatoma cells and SH-SY5Y neuroblastoma cells [19]. In these models, cells become sphere-shaped, lose adhesion and they finally become detached from the plate. Since the cytoskeleton is essential for the maintenance of cellular architecture, adhesion, migration, differentiation and transport of organelles, it is likely that toxins exert a direct or indirect effect on it. Furthermore, changes in the cytoskeleton reorganization, induced by OA and mainly mediated by inhibition of dephosphorylation, precede apoptotic processes, but they are also present in cell proliferation and cell mobility. However, it is still unclear whether the effects on the cytoskeleton are directly induced by OA, or they are a consequence of normal apoptotic processes [18].

It has been described that this group of toxins can behave as both cell death inducers and tumor promoters [20]. Del Campo et al. [2] stated that this dual effect of OA depends on the administered dose and applied exposure time, since in gastric epithelium (AGS and MKN45) and colon epithelium (Caco 2) models, sublethal doses of OA promote cell proliferation and activate oncogenic pathways [2]; however, only the effects on tumoral cells were studied herein, and nontumoral cells were excluded from it. The aforementioned background could explain that, according to our results, in more sensitive cells such as CCD 841 CoN cells, cellular death was activated, and in more resistant cells such as tumoral (Sw480 and Sw620) cells, pathways related to oncogenesis were activated by increasing, for example, the cells’ migratory capacity. Our cell migration results showed that in tumoral cells, the migration capacity was significantly increased if compared to the control, while in the nontumoral cell line, no significant differences were found. 

Toxins from DSP are potent inhibitors of PP2A and to a lesser extent of PP1, which are the most abundant protein phosphatases in cells [12]. Protein phosphatases modulate a large number of cell signaling pathways such as proliferation, differentiation and apoptosis [17]. Due to the above, in this study, PP2A activity was evaluated in the three cell lines after treatment with marine toxins. The results obtained showed significant inhibition of PP2A activity after treatments in (Sw480 and Sw620) tumoral cell lines; however, the treatment with DTX-2 did not inhibit PP2A in nontumoral cells. Due to the structural differences between OA and its analogs, it has been demonstrated that they have different affinity for the catalytic site of PP2A [21]. In this regard, Twiner et al. [9] showed that structural differences in methylation of C31 and C35 reduce the toxicity of toxins. DTX-2 shows greater affinity for PP5 than the analogs of OA and DTX-1. Likewise, when comparing the effects of DTX-2 (of natural and synthetic origin) on T lymphocytes, the toxin of natural origin turned out to be more toxic than the toxin of synthetic origin [9]. These findings, along with the fact that each cell type has a different metabolism and differential absorption of toxins, could explain the low or no inhibition of PP2A after treatments with DTX-2. However, differences in affinity for PPs do not fully explain all the effects of toxins in the OA-group [22], so there may be other cell targets for these toxins that could explain the results obtained. 

Louzao et al. [11] studied the relationship between neuropeptide Y and the exposure to OA, determining that this neuropeptide inhibits gastrointestinal mobility and secretion of water and electrolytes, mainly Cl^−^, along the intestine. Their results show that doses of OA of 50–100 nM decrease the expression of neuropeptide Y. Furthermore, OA induces a reduction in transepithelial electrical resistance (TEER) in Caco-2 cells and it is associated with a low level of neuropeptide Y which is secreted by SH-SY5Y cells [21]. Studies conducted on the effect of methyl okadaate, an artificial methyl ester derived from OA, demonstrate that it is capable of inducing reorganization of the actin cytoskeleton in human neuroblastoma cells, and also decreasing metabolic expenditure in normal and immortalized rat hepatocytes, regardless of the inhibition of PP1 and PP2A [12]. 

Moreover, the results of the evaluation of activated caspases-3/7 showed significant increases in tumor cell lines after treatment with OA and DTX-2, which correlates with the previously reported results [18]. Rossini et al. [18] reported that in HeLa and MCF-7 cells treated with 50 nM of OA for 24 h, no changes in the procaspase-8 protein expression were observed, indicating that there is no processing of these caspases by the toxin action; however, differences in isoforms of caspase-2, -3, -7 and -9 were found after treatment in HeLa cells. These results were expected since caspase-8 initiates the extrinsic pathway; therefore, it has a greater role in the apoptosis induced by TNF and Fas [18]. Increased activation of caspase-3 has been reported in normal human lung fibroblasts after incubation for 48 h with a concentration of 1–1000 nM of OA, concluding that the caspase-3 activation depends on both the used incubation time and the toxin concentration [23].

The results of the LDH release did not show significant increases in the treatments with respect to controls for the three cell lines evaluated. The same occurred for cytometry results, where no increases in the percentage PI-positive and annexin V-negative cells were observed. Both assays results, LDH release and annexin V/PI cytometry, indicated that there was no cell death by necrosis attributable to the effect of the toxins. Similar results were reported by Leira et al. [23]. They demonstrated that there were no differences in the LDH release or PI absorption in normal human lung fibroblasts after 48 h of incubation with concentrations of 1–1000 nM of OA. Davis et al. [24] demonstrated that rat kidney epithelial cells treated with 1 µM of OA were capable of excluding PI, thus demonstrating that they were able to maintain membrane integrity despite the observed morphological alterations [24].

In relation to the changes in the reported shape and motility, one of the main kinases involved in focal adhesion is FAK. FAK is involved in the remodeling of focal adhesions during the migration of cells and its activation requires phosphorylation in different tyrosine residues [25]. In endothelial cells, OA has been shown to stimulate cell mobility. Furthermore, OA produces a loss of stability of focal adhesions and the pertaining loss of keratinocyte cytoskeleton organization due to the alteration in the phosphorylation state of tyrosine residues in FAK and paxillin proteins [26]. Studies have reported that treatments with OA decrease phosphorylation of FAK in Y397, depending on the concentration and incubation time [27], contrary to what was obtained in this study. The results of protein expression of pFAK Y397 and pFAK Y576/577 showed an increase in tumoral cells with respect to the control which were statistically significant for pFAK Y397 in the Sw480 cell line after treatments with DTX-1 and DTX-2, and for pFAK Y576/577 in the Sw480 cell line after treatment with DTX-2. The nontumoral cell line did not show any significant differences in the pFAK expression, and the mRNA expression of FAK was decreased in all treatments. The increase of pFAK Y397 is correlated with an increase in the migratory capacity, since phosphorylation of FAK in this residue is relevant to the migratory process [25]. Overexpression of FAK blocks anoikis in suspended cells, which supports its role as a protector of anoikis [28]. The results obtained in the nontumoral cell line showed a decreased mRNA level of FAK, which in addition to the loss of adhesion to the plate and the morphological changes could suggest that the cause of death may be due to anoikis in this cell line, a type of programmed cell death induced by a lack of adhesion of the cell to the extracellular matrix [28]. Although anoikis and apoptosis have similar characteristics, caspases are activated later than in apoptosis during anoikis [27], which would explain the low levels of activated caspase-3/7 in CCD 841 CoN cells.

Finally, the effects of toxins on a certain cell line are dependent on the incubation time, the dose applied and the cell line used. In this study, the nontumoral cell line proved to be more sensitive to the effect of toxins, with a strong decrease in the PP2A activity only in treatment with DTX-1, with a tendency to decrease cell proliferation, and the expression of proteins associated with cell adhesion and migration. No activation of caspases-3 and -7 in the nontumoral cell line was found; it is possible that they are activated later, suggesting cell death by anoikis. However, this should be confirmed in future measurements with the evaluation of markers of the intrinsic mitochondrial pathway such as Bcl-2, which inhibits anoikis [29], or Bim, which decreases cell adhesion [28].

## 4. Conclusions

Based on the obtained results (summarized in Figure 9), we can state that a dual effect of toxins can be found since in the nontumoral cell line, a greater induction of cell death presumably by anoikis is shown. While an induction of a more aggressive phenotype is shown in the tumoral cell lines, which is characterized by an increased resistance to toxins, increased migration and activated FAK. Due to the above, this study suggests that poisoning with toxins from the OA-group, whether they reach the threshold to trigger DSP syndrome or not, could have long-term adverse effects on the digestive tract in people with non-neoplastic or neoplastic lesions. 

## 5. Materials and Methods 

### 5.1. Cell Culture

CCD 841 CoN colon epithelium cells (ATCC® CRL-1790 ™) were cultured in RPMI-1640 medium). The colorectal cancer cells SW480 (ATCC® CCL-228 ™) and SW620 (ATCC® CCL227 ™) were grown in Leibovitz’s L-15 medium (Gibco ™). Both media were supplemented with 10% fetal bovine serum (SFB; Mediatech, Manassas, VA, USA) and 1% *Penicillin / Streptomycin* (Corning Inc., Corning, NY, USA). Sw480 and Sw620 cell cultures were maintained at 37 °C without CO_2_. CCD 841 CoN cell culture were maintained at 37 °C and 5% of CO_2_, according to the ATCC recommendation.

### 5.2. Toxins

Okadaic acid (Cat. Ab120375, Abcam) was dissolved in DMSO. DTX-1 (Lot # 20151209, Certified reference materials program, Canada) and DTX-2 (Lot # 20150819, Certified reference materials program, Canada) were purchased dissolved in methanol. For the tests, the toxins were prepared in the culture media corresponding to each cell line.

### 5.3. MTT

Cells were seeded in 96-well plates at a density of 8 × 10^3^ cells / 100 µL of culture medium. After 18 hrs of adhesion, the cells were treated with the toxins in concentrations of 0, 20, 50, 100, 150 and 200 nM. After 12 h 50 μL of the MTT reagent + Locke solution (24 mM NaCl, 4mM NaHCO_3_, 5mM KCl, 10 mM HEPES, 5mM glucose, 2.3 mM CaCl_2_ 2H_2_O, 1mM MgCl_2_ 6H_2_0) were added to each well. It was incubated for 3 h at 37 °C in the dark, the MTT was discarded and 50 µL of DMSO was added to each well. Then it was incubated again for 20 min with stirring, and the absorbance at 570 nm was determined using a plate reader (Biotek Synergy HT Multi-mode microplate reader). The viability percentage was calculated: (%) = (X * 100%) / Y, where X is the absorbance of the treated cells and Y is the absorbance of the untreated cells. From this test, IC_50_ values were determined for each toxin.

### 5.4. PP2A Phosphatase 

The PP2A inhibition assay was performed using the PP2A immunoprecipitation phosphatase assay (Merck) kit, according to the manufacturer’s instructions. The cells were exposed to toxins in concentrations equivalent to IC_50_ for 12 h. 50 µg of protein was taken from each sample which was incubated with 4 µg of anti-PP2A antibody, subunit C, clone 1D6 and 25 µL of protein A agarose for 2 h at 4 °C under constant agitation. After 3 washes, the diluted phosphopeptide was added to the immunoprecipitate and incubated for 10 min at 30 °C under constant agitation. Then 25 µL of the sample was transferred to the 96-well plate, and 100 µL of malachite phosphate green detection solution was added. The samples were incubated for 15 min, and the reading was done at 650 nm on a plate reader (Biotek Synergy HT Multi-mode microplate reader).

### 5.5. Caspase-3/7 Glo

Apoptosis was evaluated using the Caspase-Glo ® 3/7 assay kit (Promega), following the manufacturer’s instructions. The cells were seeded in 96-well plates at a density of 10^4^ cells/well. After an adhesion period of 18 h, they were treated with the toxins, and the control medium was replaced with a fresh medium. After the incubation period, the medium was removed, and 50 µL of fresh medium and 50 µL of Caspase-Glo ® 3/7 reagent were added. The plate was stirred for 30 sec and subsequently incubated at room temperature for 1.5 h. This mix results in cell lysis, followed by caspase cleavage of the substrate and generation of a “glow-type” luminescent signal produced by luciferase. Luminescence is proportional to the amount of caspase activity present in the well. The luminescence measurement was then performed on a plate reader (Biotek Synergy HT Multi-mode microplate reader).

### 5.6. Annexin V /PI

The FITC annexin V kit (BD Pharmingen^TM^) was used according to the manufacturer’s instructions. The cells were seeded at a density of 10^6^ and were treated with the toxins for 12 h. After treatment, the cells were trypsinized, washed with cold PBS and centrifuged at 2500 rpm for 5 min. The supernatant was discarded and the pellet was resuspended in 1 × Binding Buffer at a concentration of 10^6^ cells/mL. 100 µL of cell suspension was transferred to a cytometry tube, and 5 µL of FITC annexin V and 10 µL of PI were added. The tubes were incubated for 15 min at room temperature in darkness. Finally, 400 µl of 1 × Binding Buffer was added to each tube, and the analysis was performed by flow cytometry.

### 5.7. LDH

To assess cytotoxicity, cells were seeded in 96-well plates at a density of 10 cells/well. The Pierce LDH citotoxicity assay kit (Cat. 88954, Thermo Fisher Scientific, Rockford, USA) was used according to the manufacturer’s instructions. After 12 h of incubation with the toxins, 10 µL of lysis Buffer (10x) was added to the controls of maximum LDH activity, and the plate was reincubated for 45 min in an oven at 37 °C and 5% CO_2_. Then 50 µL of the medium was transferred to a new well, and 50 µL of reaction mix was added. The plates were incubated for 30 min at room temperature and protected from light. Finally, 50 µL of stop solution was added and absorbance was measured at 490 and 680 nm. The cytotoxicity percentage was calculated as a percentage of the maximum activity control included in the kit.

### 5.8. Migration

The migration assays were conducted using the CytoSelect™ 96-Well cell migration and invasion kit (Cell Biolabs, San Diego, CA, USA, Cat. CBA-106-C) based on the Boyden chamber method. Briefly, cells were harvested and resuspended in serum-free medium at a density of 1 × 10^5^ cells. Next, 100 µL of the cell suspension was added to the upper chamber and 150 µL of medium supplemented with 10% fetal bovine serum to the lower chamber. The membrane separating both chambers was a polyvinyl chloride (PVC) filter with 8.0 μm pores. The chamber was incubated for 12 h at 37 °C after which the cells that had migrated to the lower surface of the filter were lysed and quantified according to the kit manufacturer’s instructions in a fluorescence plate reader at 480 nm/520 nm (Biotek Synergy HT Multi-mode microplate reader).

### 5.9. Western Blot

Western blot studies were carried out as previously described in our laboratory. Briefly, cells were grown to confluence, and then proteins were extracted using RIPA buffer and quantified by the Bradford method. For the analysis, 50 µg of protein was resolved over 10% polyacrylamide gels and electrotransferred onto a nitrocellulose membrane. The membranes were blocked with a blocking buffer for 1 h at room temperature and then incubated overnight at 4 °C with the corresponding primary antibody in blocking buffer, followed by incubation with the appropriate secondary antibody (anti-mouse HRP, 1:10000; Jackson Immunoresearch, West Grove, PA, USA, and anti-rabbit HRP, 1:10000; Jackson Immunoresearch West Grove, PA, USA) for 1 h using chemiluminescence detection. Bands were quantified using the ImageJ photo analyzing program (US National Institutes of Health, Bethesda, ML, USA), to analyze each gel. Each lane was selected; the lanes were drawn up and the area under the curve was obtained and quantified. The following primary antibodies were used: FAK Ab sampler kit (#9330, cell signaling) that includes phospho-FAK (Tyr576/577) antibody 3281, phospho-FAK (Tyr925) antibody 3284, phospho-FAK (Tyr397; D20B1) rabbit mAb 8556, FAK (D2R2E) rabbit mAb 13009 and anti-rabbit IgG, HRP-linked antibody 7074., PCNA (PC10: sc-56, Santa Cruz) and β-actin as loading control (Inmuno MP Bio, Cat. 69100).

### 5.10. qRT-PCR

Cells were grown to confluence and lysed by adding 1 ml of TRIZOL (Gibco, Cat. 15596-026) directly in the culture dish (1 ml per 3.5 cm diameter dish) and scraping. RNA was extracted and quantified using a Synergy HT Multi-detection microplate reader (BIOTEK, Winooski, VT, USA). cDNA was synthesized using the AffinityScript QPCR cDNA synthesis kit (Agilent Tech., Santa Clara, CA, USA; Cat. 600559) following the manufacturer’s protocol. qRT-PCR was performed in triplicate using the Brilliant II SYBR green QPCR master mix kit (Agilent Tech., Cat. 600828) in an Aria mix real-time PCR system (Agilent Tech., 68830A model). Data were analyzed using the AriaMX 1.0 program (Agilent Tech.). The following sets of primers were used: FAK (s): 5′-CAG GGT CCG ATT GGA AAC CA-3′, (a): 5′-CTG AAG CTT GAC ACC CTC GT-3′, PUM1 (s): 5′-CGG TCG TCC TGA GGA TAA A-3′, (a): 5′-CGT ACG TGA GGC GTG AGT AA-3′.The mRNA levels were normalized using the housekeeping gene PUM1.

## Figures and Tables

**Figure 1 toxins-12-00179-f001:**
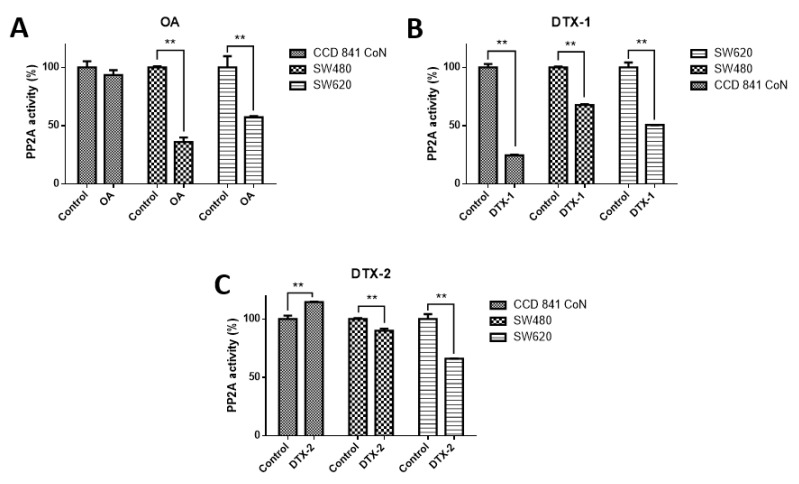
Type 2A protein phosphatase (PP2A) activity in normal and colon tumoral cell lines. The CCD 841 CoN, Sw480 and Sw620 cell lines were treated for 12 h with okadaic acid (OA) toxins (**A**), dinophysistoxin-1 (DTX-1), (**B**) and dinophysistoxin-2 (DTX-2), (**C**) at concentrations equivalent to IC_50_ obtained by the MTT assay, subsequently, the PP2A immunoprecipitation phosphatase assay kit was used (Merck) according to the manufacturer’s instructions. The bars indicate the average value with respect to the control ± standard error. Significant differences were calculated using the student’s *t*-test. *n* = 3, ** *p* < 0.01.

**Figure 2 toxins-12-00179-f002:**
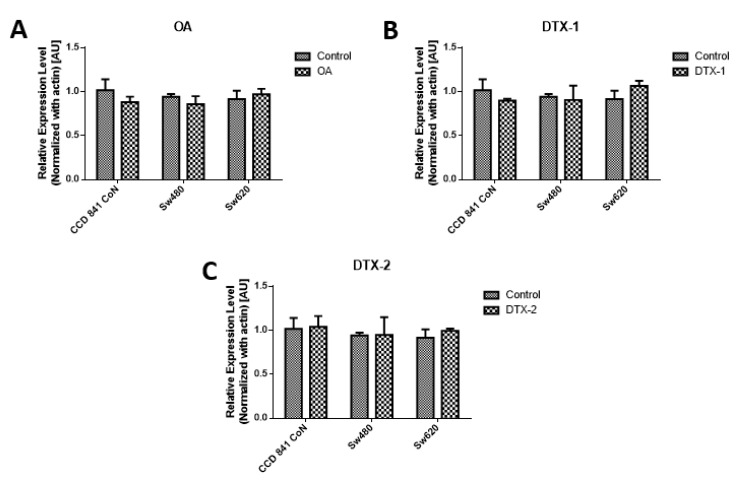
Assessment of proliferating cell nuclear antigen (PCNA). Results of PCNA expression of Con841 (control with no toxin, OA, DTX-1, DTX-2), Sw480 (control with no toxin, OA, DTX-1, DTX-2) and Sw620 (control with no toxin, OA, DTX-1, DTX-2) cell lines. Cells were treated with concentrations equivalent to IC_50_ obtained using the MTT assay. (**A**) The relative expression level of PCNA in cells treated with OA (*n* = 3). (**B**) The relative expression level of PCNA in cells treated with DTX-1 (*n* = 3). (**C**) The relative expression level of PCNA in cells treated with DTX-2 (*n* = 3). Bars indicate average ± standard error.

**Figure 3 toxins-12-00179-f003:**
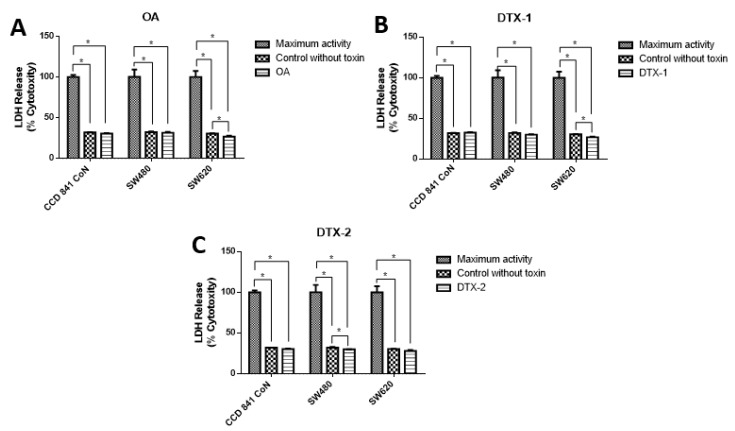
Cytotoxicity was determined using lactate dehydrogenase LDH assay. Cells were treated with OA (**A**), DTX-1 (**B**) and DTX-2 (**C**) with concentrations equivalent to IC_50_ for each toxin obtained using the MTT assay. Then, the LDH was measured in the culture medium using the Pierce LDH cytotoxicity assay kit according to the manufacturer’s instructions. The bars indicate the average value with respect to the maximum LDH activity control ± standard error. Significant differences were calculated using the student’s *t*-test. (*n* = 4), * *p* < 0.05.

**Figure 4 toxins-12-00179-f004:**
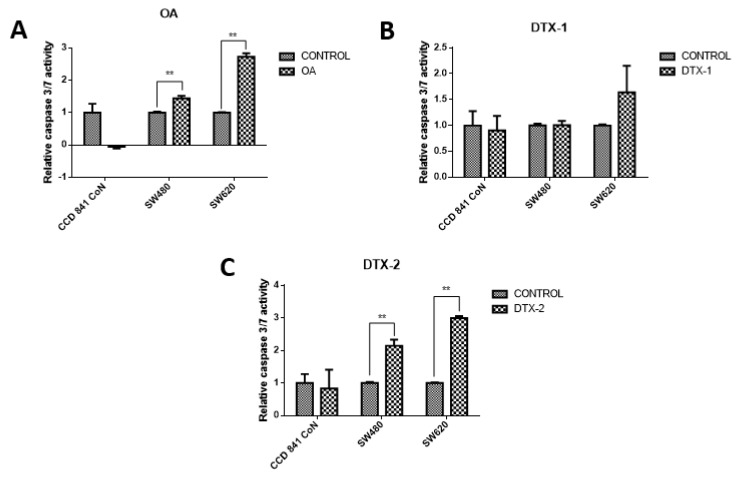
Evaluation of the caspase-3/7 activity. CCD 841 CoN, Sw480 and Sw620 cell lines were incubated for 12 h with OA (**A**), DTX-1 (**B**) and DTX-2 (**C**) with concentrations equivalent to IC_50_ for each toxin obtained using the MTT assay. Then, cell apoptosis was measured by caspase-3/7 assay (Promega). The values are expressed as a percentage of the control. Bars indicate average ± standard error. Significant differences were calculated using the student’s *t*-test. (*n* = 3), ** *p* < 0.001.

**Figure 5 toxins-12-00179-f005:**
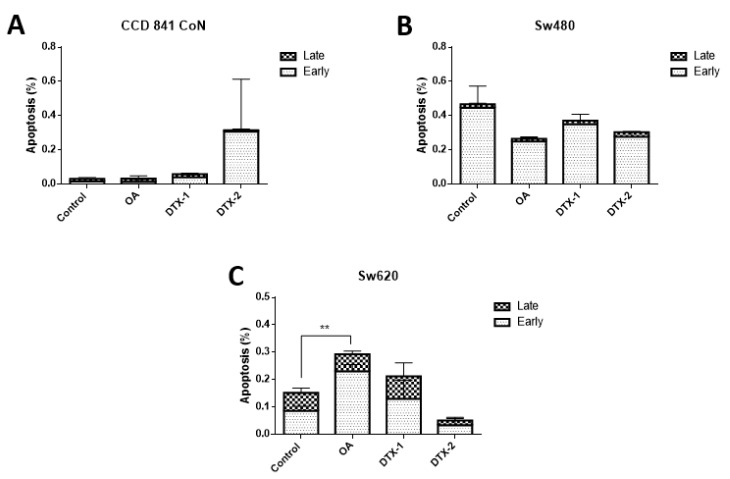
Evaluation of apoptosis by annexin V/PI assay and analysis by flow cytometry. CCD 841 CoN (**A**), Sw480 (**B**) and Sw620 (**C**) cells were treated for 12 h with OA, DTX-1 and DTX-2 in concentrations equal to IC_50_ obtained using the MTT assay. The population of early apoptosis is characterized by annexin V (+) and PI (−); the population of late apoptosis is characterized by annexin V (+) and PI (+). Bars indicate mean ± standard error. Significant differences were calculated using the student’s *t*-test. (*n* = 3), ** *p* < 0.01.

**Figure 6 toxins-12-00179-f006:**
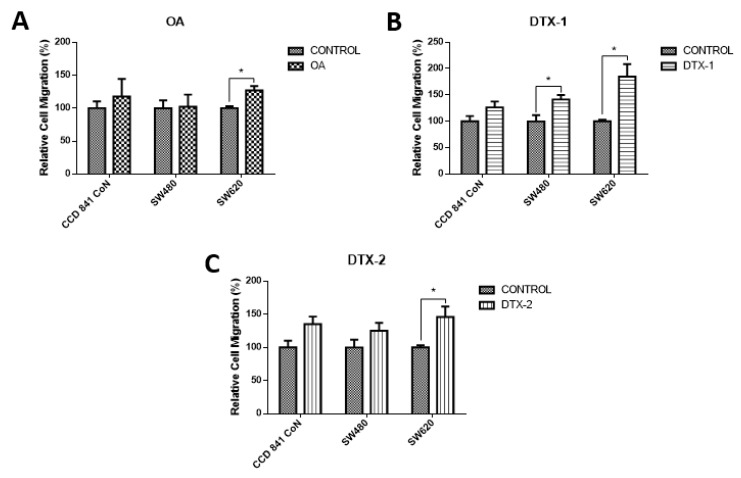
Evaluation of cell migration. Transwell chambers were used to evaluate the migratory capacity of the cells. Cells were incubated for 12 h with OA (**A**), DTX-1 (**B**) and DTX-2 (**C**) toxins with concentrations equivalent to IC_50_ obtained by the MTT assay; then, the Cell Biolabs Cyto-Select^TM^ (Cell Biolabs) assay kit with 96-well and 8 µm pore size was used. Bars indicate average ± standard error. Significant differences were calculated using the student’s *t*-test. Student. (*n* = 4), * *p* < 0.05.

**Figure 7 toxins-12-00179-f007:**
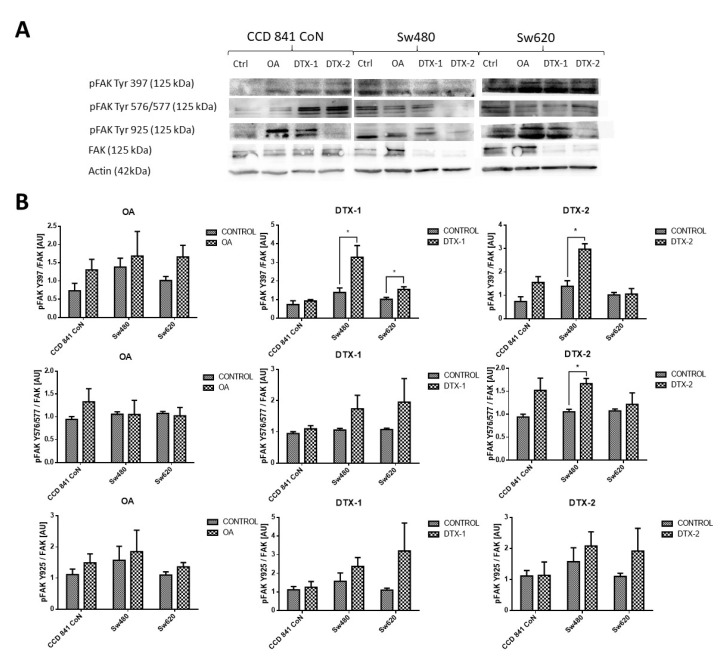
Antigen evaluation of FAK and pFAK. (**A**). Results of Western blot for FAK and pFAK Tyr 576/577, pFAK Tyr 925 and pFAK Tyr 397 in CCD 841 CoN, Sw480 and Sw620 (control with no toxin, OA, DTX-1, DTX-2) cell lines. Polyacrylamide gel (10 %; *n* = 3). (**B**). The relative quantification of pFAK Y397/FAK, pFAK Y576/577/ FAK and pFAK Y925/FAK for each cell line after treatment with OA, DTX-1 and DTX-2. Bars indicate average ± standard Error. Significant differences were calculated using the student’s *t*-test. * *p* < 0.05.

**Figure 8 toxins-12-00179-f008:**
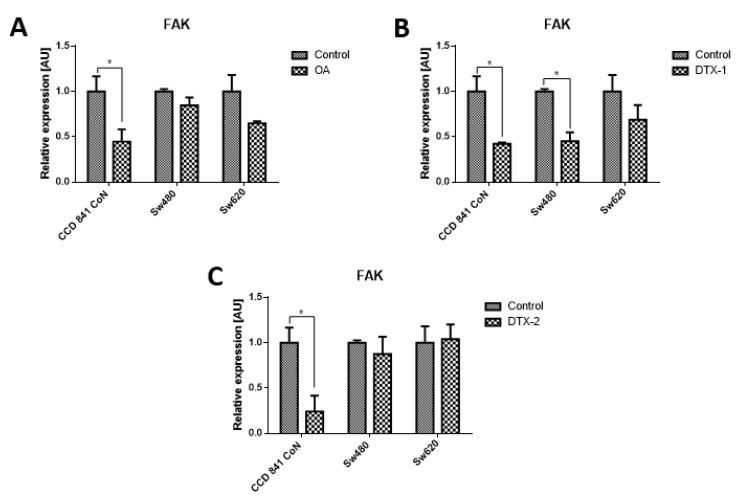
Relative expression level of mRNA of FAK. CCD 841 CoN, Sw480 and Sw620 cells were treated for 12 h with OA (**A**), DTX-1 (**B**) and DTX-2 (**C**) toxins in concentrations equivalent to IC50 obtained by the MTT assay. Quantification was performed using RT-qPCR. Bars indicate average ± standard error. Significant differences were calculated using the student’s *t*-test. (*n* = 6), * *p* < 0.05.

**Figure 9 toxins-12-00179-f009:**
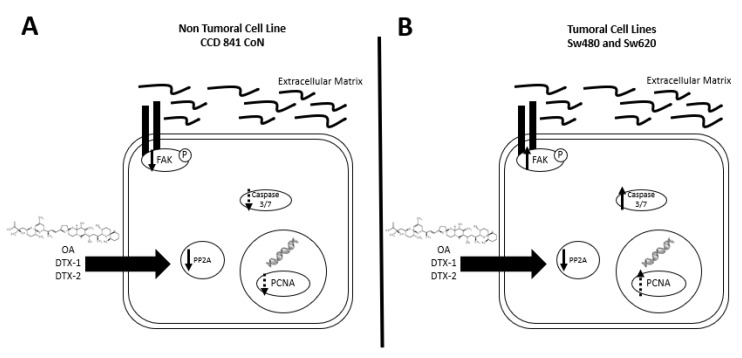
Summary of results. (**A**) In the CCD 841 CoN cell line, a tendency towards decreased expression of PCNA, decreased PP2A activity and a decreased pFAK with no activation of caspases-3/7 are found after incubation with OA, DTX-1 and DTX-2. (**B**) In Sw480 and Sw620 cell lines, a decreased PP2A, a tendency towards an increased PCNA, increased expression of pFAK and activation of caspases-3/7 were observed, after incubation with OA, DTX-1 and DTX-2.

**Table 1 toxins-12-00179-t001:** IC_50_ values for each toxin obtained from the MTT assay.

Cell Line	IC_50_ OA (nM)	95% IC_50_	IC_50_ DTX-1 (nM)	(95%) IC_50_	IC_50_ DTX-2 (nM)	(95%) IC_50_	
CCD 841 CoN	54.4	36.58–80.84	43.5	29.71–63.78	81.2	70.28–93.75
Sw480	89.1	65.31–121.6	113.8	83.86–154.3	187.2	163.6–214.2
Sw620	137.8	99.96–189.9	192.9	150.6–247.0	202.9	179.6–229.2

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
