# Peer review of "Toxins of Okadaic Acid-Group Increase Malignant Properties in Cells of Colon Cancer"

_toxins, 2020, doi:10.3390/toxins12030179_

Round 1
Reviewer 1 Report
The manuscript describes and compares the effect of DSP toxins in survival and mobility of colon cancer and healthy colon epithelial cell lines. The study addresses an important and current issue and presents novel results in both DSP toxin and colon cancer research. The research is well structured, presented with care and in detail. The abstract gives a concise overall of the work and highlights the major results. The Introduction presents to the reader an overview of the state of art but will benefit from simple rephrases. The references cited are recent and appropriated. The discussion places the research in a broader context and relates it to other studies.
For the reasons mentioned above, the manuscript is worth being published in the Journal after minor revisions.
Specific comments:
Introduction:
Line 26 – The first sentence is a poor start for such a fine manuscript. It barely describes HABs and it does not give a connection with DSP toxins and the producer organisms. It should be rephrased.
Line 32 – The sentence “Okadaic acid is the main analogue of DSP. Other toxins are analogues of OA:” is confusing should be rephrased.
Line 63 – Reference number missing for Valdiglesias et al. (2013) and others throughout the text.
Line 160 – The figure caption is not formatted.
Author Response
Reviewer 1
The manuscript describes and compares the effect of DSP toxins in survival and mobility of colon cancer and healthy colon epithelial cell lines. The study addresses an important and current issue and presents novel results in both DSP toxin and colon cancer research. The research is well structured, presented with care and in detail. The abstract gives a concise overall of the work and highlights the major results. The Introduction presents to the reader an overview of the state of art but will benefit from simple rephrases. The references cited are recent and appropriated. The discussion places the research in a broader context and relates it to other studies.
For the reasons mentioned above, the manuscript is worth being published in the Journal after minor revisions.
Specific comments:
Introduction:
Line 26 – The first sentence is a poor start for such a fine manuscript. It barely describes HABs and it does not give a connection with DSP toxins and the producer organisms. It should be rephrased.
ANS: Done.
Line 32 – The sentence “Okadaic acid is the main analogue of DSP. Other toxins are analogues of OA:” is confusing should be rephrased.
ANS: Done . Okadaic acid is the main analogue of DSP toxins. Other toxins of this group are: dinophysistoxin-1 (DTX-1), dinophysistoxin-2 (DTX-2).
Line 63 – Reference number missing for Valdiglesias et al. (2013) and others throughout the text. ANS: Done.
Line 160 – The figure caption is not formatted.
ANS: Done.
Reviewer 2 Report
In the manuscript the authors try to determine how the toxins of the OA-Group increase the malignant properties of cancer cell lines. However, the authors did a series of experiments to study cell survival and mobility that are not actually a direct link of the malignant or oncogenic capacities of the cancer cell lines. And in some cases the kind of experiments performed are not adequate for the questions that the authors want to answer. For instance, study PCNA levels is not the best way (or standardised method) to measure proliferating cells.
Most of the raw data is not presented, for instance the cell viability assays are only summarised in Table 1, but not the actual curves are shown. The PCNA western blot are now shown (although as mentioned above are not the best way to measure proliferating cells).
The western blots to assess pFAK are cut so the comparison between samples are not direct. Right now, it is not possible to know if different exposure times are used or different western blot have been used for comparison. All the samples should be run in the same gel an presented as a unique western blot.
The results section is extremely poorly written. There is no introduction to explain why the specific assays is performed and what they want to achieve in each assay. No explanation of how the experiments were done, and only a brief description of the results are included.
Moreover, only one non-tumoral cell line is used and more control cell lines are needed to compare the diversity of the results the authors presented.
In general, I will suggest the authors to concentrate in the most important message they want to give an focus on the best results that support them. Right now, some contradictory results does not allow to clearly see their message.
Author Response
Dear Reviewer 2 "Please see the attachment".

Reviewer 3 Report
Please find attached my Review Report.

Author Response
Reviewer 3.
Brief Summary: In this manuscript, the authors investigate the effects of Diarrhetic Shellfish Poisoning toxins Okadaic acid and its analogues on cell survival and motility in non-tumoral and tumoral colon cancer cell lines. Toxins induced aggressive phenotype changes in tumoral cell lines, with increased resistance to toxins, increased FAK activation and, increased migration.
Specific Comments:
Manuscript Title:
Line 2: Expand OA to Okadaic Acid in the title.
ANS: Done.
Introduction:
Line 32: Okadaic acid is the main analogue of DSP toxins
ANS: Done .Okadaic acid is the main analogue of DSP toxins. Other toxins of this group are: dinophysistoxin-1 (DTX-1), dinophysistoxin-2 (DTX-2).
Line 37: Change type 2A to type 1
ANS: Done. The main molecular targets of this group of toxins are type 1 protein phosphatase (PP1) and type 2A protein phosphatase (PP2A).
Line 38: type 2A protein phosphatase (PP2A)
ANS: Done.
Line 33-57: Modify the sentence- At high doses – equivalent to acute poisoning- the intestinal barrier function is lost and toxins pass through the bloodstream to other tissues, producing ulcers, hyperplasia, necrosis, and atrophy.
ANS: Suggestion accepted. Although, it has been determined that after high concentrations of OA, corresponding to acute poisoning, toxins pass through the bloodstream to other tissues, producing ulcers, hyperplasia, necrosis and atrophy.
Line 58: remove ‘In this regard’
ANS: Done.
Line 59: Remove ’That’s how’
ANS: Done.
Results
Line 84: Change the sentence to activity was significantly decreased only in tumoral cell lines.
ANS: Done.
Line 84: What is the percentage of the decrease in OA treated cell lines?
ANS: Done (CCD 841 CoN: 93.4%; Sw480: 36%; Sw620: 57.2%).
Line 90: Fig 1A; change AO to OA
ANS: Done.
Line 97: Here it is mentioned that there are no significant variations in PCNA protein expression. But this contradicts the Summary of results described in Figure 9 (line 295-296), where it is mentioned that in CCD 841 CoN cell line, there is a tendency towards decreased expression of PCNA and in Sw480 and Sw620 cell lines, there is tendency towards an increased PCNA. Please clarify this.
ANS: There is a tendency to decrease the expression of PCNA in CCD841 CoN cells. In contrast in Sw620 cells, a tendency to increase PCNA expression is observed. This suggests a differential proliferation response depending on the type of cell line, tumor or non-tumor used, although changes are not statistically significant.
Line 100: In Figure 2A, change AO to OA
ANS: Done.
Line 102-104: Please correct the description of Figure 2A and 2B.
ANS: Done.
Line 102: Remove A
ANS: Done.
Line 103: Replace B with A
ANS: Done.
Line 104: Replace C with B
ANS: Done.
Line 111-112: Statistical difference is not very convincing based on the figure. It would be better if actual % Cytotoxicity between the groups were mentioned.
ANS: Done . OA: CCD 841 CoN 30.4%, Sw480 31.5%, Sw620 26.5%; DTX-1: CCD 841 CoN 32.5%, Sw480 30%, Sw620 26.8%; DTX-2: CCD 841 CoN 30.3%, Sw480 29.9%, Sw620 28.1%.
Line 116: In Figure 3A, change AO to OA
ANS: Done.
Line 123: In figure 4A, change AO to OA
ANS: Done.
Line 123: In Figure 4A, why is the Caspase 3/7 activity is below 0 in CCD 841 CoN when treated with OA?
ANS: Done. The control values were normalized to 1. The luminescence of the CCD 841 CON cell line treated with OA was lower than the control.
Line 135: In Figure 5, change AO to OA
ANS: Done.
Line 146: In Figure 6A, change AO to OA
ANS: Done.
Line 160: Figure 7 is out of focus. Please replace with a figure with good resolution.
ANS: Done.
Line 160: In Figure 7, change AO to OA
ANS: Done.
Line 160: In Figure 7, what is the unit of protein expression level on X-axis?
ANS: Suggestion accepted and changes made. Y axis represents arbitrary units, X axis represents cell lines.
Line 161-163: There is no need to mention the treatment conditions (Control with no
toxin, OA, DTX-1, DTX-2) for each cell line.
ANS: Done.
Line 169: In Figure 8A, change AO to OA
ANS: Done.
Discussion:
Line 175: Remove ‘from each other’.
ANS: Done.
Line 180: Replace ‘the first ones’ with non tumoral cell line CCD 841 CoN
ANS: Done.
Line 181: Replace ‘if ‘ compared with when compared with
ANS: Done.
Line 247: Discuss decreased LDH activity mentioned in RESULTS section (line 111-
112).
ANS: Suggestion accepted and phrase replaced. Both assays’ results, LDH release and Annexin V/PI cytometry, indicate that there is no cell death by necrosis attributable to the effect of the toxins.
Materials and methods:
Line 306: Please provide cell culture conditions.
ANS: Suggestion accepted and phrase replaced. Sw480 and Sw620 cell cultures were maintained at 37°C without CO2. CCD 841 CoN cell culture were maintained at 37°C and 5% CO2, according to the ATCC recommendation.
Line 313: density of 8X 103 cells
ANS: Done.
Line 322: Change fosfatase to phosphatase.
ANS: Done.
Line 325: What is the control treated with?
ANS: Suggestion accepted and phrase replaced. Medium of controls were replaced with fresh medium.
Line 325: Please mention how the protein was extracted
ANS: Suggestion accepted and phrase included (Line 370: This mix results in cell lysis, followed by caspase cleavage of the substrate and generation of a “glow-type” luminescent signal, produced by luciferase. Luminescence is proportional to the amount of caspase activity present in the well.
Line 342: 106 cells/ well
ANS: Done.
Line 370: (18).
ANS: Done.
Line 375: What is the secondary antibody used in the Western Blot? Also, Please provide the name of the manufacturer.
ANS: Done . Anti-mouse HRP (1:10000, Jackson Immunoresearch, West Grove, PA, USA), and anti-rabbit HRP (1:10000, Jackson Immunoresearch West Grove, PA, USA).
Line 377: Describe how the relative protein expression is measured using ImageJ software
ANS: Suggestion accepted and phrase included. Line 414: to analyse the bands of the respective gels, each lane was selected and then with the use of the Image Software, the area under the curve of each condition was obtained. It is then normalised with actin as a control.
Line 378: Please provide the manufacturer’s name for pFAK (pFAK Tyr 397 and pFAK Tyr 576/577, and pFAK Tyr 925) primary antibody
ANS: Done. That includes Phospho-FAK (Tyr576/577) Antibody 3281, Phospho-FAK (Tyr925) Antibody 3284, Phospho-FAK (Tyr397) (D20B1) Rabbit mAb 8556, FAK (D2R2E) Rabbit mAb 13009 and Anti-rabbit IgG, HRP-linked Antibody 7074.
Line 398: In Figure A1, change AO to OA
ANS: Done.
Text editing:
Please carefully proofread to fix all the punctuation errors.
Round 2
Reviewer 2 Report
The authors provided an updated version of the manuscript. Although I do not consider that the research design is appropriate for some experiments (like the cell proliferation) already mentioned in my previous revision, right know the manuscript and the data are overall better fitting with the authors's research conclusions.